# CAR-T Therapy Beyond B-Cell Hematological Malignancies

**DOI:** 10.3390/cells14010041

**Published:** 2025-01-03

**Authors:** Martina Canichella, Paolo de Fabritiis

**Affiliations:** 1Hematology, St. Eugenio Hospital, ASL Roma2, 00144 Rome, Italy; paolo.defabritiis@aslroma2.it; 2Department of Biomedicina e Prevenzione, Tor Vergata University, 00133 Rome, Italy

**Keywords:** CAR-T in hematology, CAR-T in solid neoplasia, infectious and autoimmune disease, tumor microenvironment, tumor immunosuppression

## Abstract

Despite the advances of CAR-T cells in certain hematological malignancies, mostly from B-cell derivations such as non-Hodgkin lymphomas, acute lymphoblastic leukemia and multiple myeloma, a significant portion of other hematological and non-hematological pathologies can benefit from this innovative treatment, as the results of clinical studies are demonstrating. The clinical application of CAR-T in the setting of acute T-lymphoid leukemia, acute myeloid leukemia, solid tumors, autoimmune diseases and infections has encountered limitations that are different from those of hematological B-cell diseases. To overcome these restrictions, strategies based on different molecular engineering platforms have been devised and will be illustrated below. The aim of this manuscript is to provide an overview of the CAR-T application in pathologies other than those currently treated, highlighting both the limits and results obtained with these settings.

## 1. Introduction

The remarkable success of chimeric antigen receptor (CAR)-T cell in treating specific B-cell hematological diseases, such as non-Hodgkin lymphomas, B-cell acute lymphoblastic leukemia (B-ALL) and multiple myeloma (MM), led to extending its application to other fields [1,2,3,4,5]. One of these is relapsed/refractory (R/R) T cell acute lymphoblastic leukemia (T-ALL), a disease with poor prognosis and limited therapeutic options, which well represents a condition in which CAR-T has shown promising results. Moreover, CAR-T strategies have been expanded to solid tumors as well as non-oncological conditions, including autoimmune diseases and chronic infections. The architecture of CAR consists of four distinct components, each serving a specific role. These include the extracellular antigen recognition domain, constituted by a single-chain variable fragment (scFv); the hinge domain, which provides flexibility and facilitates correct linking with the target antigen; the transmembrane domain; and the intracellular domain [6]. To date, five generations of CARs have been developed, primarily distinguished by variations in the design of the intracellular signaling region [7]. CAR-T cell therapy faces two well-recognized toxicities, namely cytokine release syndrome (CRS) and immune effector cell-associated neurotoxicity syndrome (ICANS) [8], in these new contexts as well. CRS, the most common CAR-T-related toxicity, arises from an excessive inflammatory response driven by the hyperactivation of effector cells and the release of various cytokines. Typically, CRS manifests within the initial weeks following CAR-T infusion and presents with a range of symptoms. Knowledge of the underlying pathological mechanisms has forced clinical protocols to classify and manage manifestations of CRS. Tocilizumab, a monoclonal antibody targeting the IL-6 receptor, has proven effective in mitigating CRS-associated toxicities and is widely used for grade 2 CRS [9,10]. In contrast, ICANS, a central nervous system (CNS)-related complication, is less understood from a pathophysiological perspective. While ICANS frequently occurs alongside or after CRS, it can also arise independently. Early intervention with dexamethasone is the standard first-line treatment for ICANS. Additionally, anakinra, a recombinant IL-1 receptor antagonist capable of crossing the blood–brain barrier, has demonstrated effectiveness in targeting IL-1β within the CNS to alleviate ICANS symptoms.

Advances in managing CRS and ICANS have significantly improved the safety profile of CAR-T cell therapy, reducing associated mortality. However, CAR-T application in AML, T-ALL and solid tumors presents additional challenges, compared with those treated in B-cell hematological disease, which we summarize in Table 1. The aim of this review is to provide an overview aimed at offering insights into the application of CAR-T cells in different settings. To this end, we have sought to summarize the rationale underlying the application of CAR-T in three major fields: the hematological malignancies beyond B-cell neoplasms (AML and T-ALL), solid tumors and chronic diseases (including autoimmune disorders and chronic infections), shown in Figure 1. Within each of these three main sections, we have highlighted the most significant evidence, emphasizing the promising results that are expected to be further validated in the near future.

## 2. CAR-T in Acute Myeloid Leukemia

While CAR-T cell therapy has shown remarkable success in certain B-cell lymphoproliferative disorders, its clinical effectiveness in acute myeloid leukemia (AML) remains limited. A major obstacle is the lack of a specific antigen that meets the necessary criteria for CAR targeting, such as high specificity, stable expression and avoidance of normal hematopoietic stem cells (HSCs) and healthy tissues to reduce off-target effects. Currently, CAR-T therapy in AML is predominantly being investigated in clinical trials involving patients with primary refractory disease or post-transplant relapse. Recent advancements in CAR-T therapies targeting CD33, CD123, CLL-1 and FLT3 have demonstrated significant progress in both preclinical and clinical settings. We summarize in Table 2 the ongoing clinical trials with CAR-T AML against these four main targets. CD33, a transmembrane receptor in the sialic acid-binding immunoglobulin-like lectin (Siglec) family, is expressed in 90% of AML cells, including leukemic stem cells and blasts [11]. Early preclinical studies of CD33-targeted CAR-T cells showed potent anti-leukemic activity but encountered challenges such as CAR-T cell exhaustion and myelosuppression [12,13]. To overcome these issues, researchers have developed modified constructs, including CD33-CAR-T cells with transient CD33 expression or alternative co-stimulatory domains and antigen-binding regions [14,15]. Novel approaches like Universal CARs (UniCARs) have also been employed to mitigate off-target effects, particularly myelosuppression. These innovations underscore ongoing efforts to enhance the efficacy and safety of CD33-CAR-T therapies for AML treatment.

CD123, the alpha subunit of the interleukin-3 receptor (IL-3Rα), is overexpressed in AML and other hematological malignancies, but is also found on normal HSCs [16]. Initial preclinical studies revealed robust anti-leukemic activity with CD123-targeted CAR-T cells but raised concerns regarding significant myeloablation. Several strategies have been developed to address this issue, including transient RNA-electroporated CAR-T cells targeting CD123 (RNA-CART123), CAR-T cell depletion using alemtuzumab and the rituximab-mediated depletion of CAR-T cells co-engineered to express CD20 [17]. Additionally, UniCAR-T cells targeting CD123 have demonstrated strong anti-leukemic effects while minimizing hematotoxicity. Reversed CAR (RevCAR) systems, which differ from UniCARs in the configuration of their binding domains, have also been explored [18]. Promising results have emerged from allogeneic CAR-T cells with gene-editing platforms to eliminate TCR α/β and reduce the risk of graft-versus-host disease (GvHD). Although clinical data remain limited, some studies have shown complete remission in heavily pretreated patients using donor-derived anti-CD123 CAR-T cells, with ongoing trials further evaluating the safety and efficacy of these therapies [18]. CLL-1, also known as C-type lectin-like molecule-1, is a type II transmembrane glycoprotein expressed on approximately 92% of AML blasts and leukemia stem cells (LSCs) while being absent on normal CD34+CD38- hematopoietic stem cells (HSCs) [19,20]. This selective expression makes CLL-1 an attractive target for AML treatment. Initial preclinical studies using CAR-T cells targeting CLL-1 demonstrated potent cytotoxicity against AML models but raised concerns about toxicity [21,22]. The first clinical application of CLL-1-CAR-T therapy involved a pediatric patient with secondary AML, who achieved complete remission with MRD negativity after treatment, despite requiring a second infusion of CAR-T cells.

FLT3, a membrane-bound glycosylated protein in the class III receptor tyrosine kinase family, plays a critical role in normal hematopoiesis and is overexpressed in 54–92% of AML cases [23,24]. Mutations in the FLT3 gene, such as internal tandem duplications (ITD) and tyrosine kinase domain (TKD) mutations, are prevalent in AML [23,24]. Second-generation CAR-T cells targeting FLT3 with a 4-1BB co-stimulatory domain and FLT3 ligand (FLT3L) antigen-binding regions have shown promising cytotoxicity against leukemic cells with FLT3-ITD mutations while sparing wild-type FLT3-expressing cells [25]. Combining anti-FLT3 CAR-T cells with FLT3 inhibitors like crenolanib or gilteritinib has been investigated but resulted in higher toxicity. To address these concerns, modified CAR-T constructs with safety switches, such as anti-FLT3 CAR-R2 T cells, have been developed. These include mimotopes of rituximab inserted within the CAR construct, allowing CAR-T cell depletion through rituximab administration [26]. This approach balances therapeutic efficacy and safety by enabling bone marrow recovery and reducing hematotoxicity [26].

These advancements in CAR-T therapy highlight significant progress in overcoming the challenges associated with targeting AML-specific antigens, offering hope for more effective and safer treatment strategies in the future.

## 3. CAR-T in T-Cell Acute Lymphoblastic Leukemia

The prognosis for patients with relapsed/refractory (R/R) T cell acute lymphoblastic leukemia (T-ALL) remains poor due to the acquisition of chemoresistance and the limited availability of effective treatments. Targeted therapies, including CAR-T cell therapy, are currently under active investigation and clinical application; however, several challenges must be addressed, including the fratricidal effect caused by the co-expression of T cell targets on the CAR-T cells themselves, T cell aplasia and the lack of a highly specific target for T cell malignancies. Despite these obstacles, ongoing clinical studies, albeit preliminary studies, have shown promising results (Table 3).

CD7, a transmembrane glycoprotein normally expressed on T cells, natural killer (NK) cells and their precursors, is also found in T cell leukemia and lymphoma, positioning it as a promising target for CAR-T therapy. Approaches to suppress CD7 expression on CAR-T cells include CRISPR-Cas9-mediated gene editing, natural selection methods and retention within the endoplasmic reticulum [27].

In the pediatric setting, Campana et al. ideated an anti-CD7 CAR construct incorporating a protein expression blocker (PEBL) to prevent CAR-T cells from lysing themselves [28]. In a study of 17 patients with R/R T-ALL treated with this construct, 16 achieved minimal residual disease MRD-negative status within one month, with 11 experiencing sustained and persistent remission. Similarly, in a phase I clinical trial involving 12 adult patients with R/R T-ALL, Hu et al. demonstrated the safety and efficacy of the PEBL-based anti-CD7 CAR-T therapy, leading to the initiation of a phase II trial. These remarkable clinical responses have generated significant interest, resulting in approximately 17 ongoing trials actively recruiting participants [29,30].

Another promising target for CAR-T therapy is CD4, a molecule universally expressed on peripheral T cell lymphoma cells. Preclinical studies have highlighted the potential of CD4-CAR-T cells as a viable therapeutic strategy for R/R T-ALL [31].

In the subset of cortical T-ALL, the most common T-ALL subtype, CD1a has emerged as a valid target for CAR-T therapy. Preclinical studies have shown the efficacy of anti-CD1a CAR-T cells, which has led to the initiation of a phase II clinical trial (NCT05745181) [32].

Indeed, CD99, a glycosylated transmembrane protein highly expressed in most T-ALL cases, has multiple biological roles, including regulating processes such as cell death, differentiation, adhesion, migration and intracellular protein trafficking. These characteristics make CD99 a promising CAR-T target, currently under investigation in two multi-institutional phase I trials (ChiCTR2100046764, ChiCTR2000033989) [33].

## 4. CAR-T in Solid Tumors

Similarly to the advancements achieved in hematological malignancies, precision medicine has significantly improved the outcomes of certain solid tumors using targeted therapies. In the last decade, the success of CAR-T cell therapy in treating specific B-cell malignancies led to its application in solid tumors. Various target molecules have been explored, yielding promising preliminary results [34]; however, the treatment of solid tumors presents unique challenges distinct from those encountered in hematological diseases. These include the lack of highly specific targets on tumor cells, the hostile tumor microenvironment (TME) and the limited infiltration of CAR-T cells into the tumor mass. To overcome these obstacles, several innovative CAR-T constructs have been developed.

This section first outlines the limitations of CAR-T cell therapy in non-hematological malignancies, followed by a discussion of the most promising targets and their results in solid tumor therapy.

### 4.1. Barriers to the Effectiveness of CAR-T Cell Therapy in Solid Tumors

Three major and unique obstacles are encountered in the use of CAR-T cells for the treatment of solid tumors. First, the infiltration of CAR-T cells into the tumor mass is hindered by a complex network of chemokines and their receptors. Second, the immunosuppressive tumor microenvironment protects the tumor while simultaneously promoting its survival and growth. Lastly, the reduced persistence of CAR-T cells may compromise their efficacy over time. Each of these challenges must be carefully addressed to enhance the therapeutic potential of CAR-T cell therapy in solid tumors [35].

#### 4.1.1. Tumor Infiltration

The efficacy of CAR-T cell therapy in neoplasia requires successful access to the tumor mass, which is dependent on the interaction between chemokines secreted by neoplastic cells and corresponding chemokine receptors (CCRs) on CAR-T cells. To address this challenge, CAR-T cells engineered to express specific CCRs have been developed to enhance infiltration into the tumor mass. Promising results have been observed with CXCR2 CAR-T cells against melanoma, CCR2b CAR-T cells against mesothelioma and GD2-CAR-T cells against neuroblastoma [36,37]. Additionally, CAR-T cells expressing CXCR1 or CXCR2, which bind IL-8—a chemokine implicated in tumorigenesis within the tumor microenvironment—have demonstrated efficacy in targeting prostate, ovarian, melanoma and colon cancer cells.

An additional significant obstacle to CAR-T cell infiltration is abnormal tumor vascularization, which is critical for tumor nutrition and cell circulation. CAR-T cells targeting key receptors involved in angiogenesis have been explored to overcome this barrier [38]. For instance, CAR-T cells against VEGFR-2 have shown promise in metastatic melanoma, while those targeting αvβ6 integrin have demonstrated encouraging responses in cholangiocarcinoma, and ovarian and breast cancers.

To further enhance CAR-T cell migration and infiltration, innovative strategies, such as engineering CAR-T cells to secrete matrix-degrading enzymes or antigens, like fibroblast activation protein, have been investigated. These modifications help disrupt the dense extracellular matrix that physically impedes CAR-T cells. Additionally, CAR-T cells designed to degrade the protective glycocalyx, often referred to as the “sugar coat” enveloping tumor cells, has emerged as a promising target. This innovative molecular “torpedo” strategy disrupts the sugar barrier, enabling CAR-T cells to penetrate and effectively eliminate solid tumors [39].

#### 4.1.2. Immunosuppressive Tumor Microenvironment

The tumor microenvironment (TME) is a complex structure characterized by hypoxic regions and a network of immunosuppressive cells which support tumor growth, including regulatory T cells (Tregs), tumor-associated macrophages (TAMs) and myeloid-derived suppressor cells (MDSCs) [40]. Targeting these cells with CAR-T might be a potential strategy to achieve therapeutic efficacy. Armored CAR-T cells, engineered to secrete immunostimulatory IL-12, IL-18 and IL-15 cytokines, have been developed with the aim of modulating the immunological microenvironment by recruiting endogenous immune cells, such as memory T cells and central memory T cells, thereby improving CAR-T cell survival.

Remarkably, IL-18-secreting CAR-T cells have shown improved anti-tumor activity by stimulating the activation of NK cells and M1-polarized macrophages, which enhance inflammation and promote anti-tumor immune responses [41].

Given that Tregs, MDSCs and M2-polarized macrophages are primary orchestrators of immunosuppression in the TME, various CAR-T strategies have been devised to target these cells. Tregs regulate the TME through the release of TGF-β, making it a potential target [42]. However, since TGF-β receptors are also expressed on CAR-T cells, CRISPR/Cas9 technology has been employed to knock out the TGF-β receptor gene in CAR-T cells, preventing their depletion and enhancing their effectiveness [43].

MDSCs inhibit T cell immune activity through mechanisms like the release of nitric oxide (NO) and reactive oxygen species (ROS). To mitigate the oxidative damage caused by NO, combination strategies such as the administration of all-trans retinoic acid (ATRA) have been explored [44].

#### 4.1.3. T Cell Exhaustion

CAR-T cell exhaustion represents one of the primary limitations of the CAR-T therapy strategy. To address this issue, second-generation CAR-T cells have been developed, incorporating co-stimulatory molecules such as 4-1BB or CD28 [45]. These second-generation CAR-T cells are currently approved for treating hematological malignancies of B-cell origin. Several studies have demonstrated that CD28 intracellular domains enhance effector functions, whereas 4-1BB intracellular domains promote greater CAR-T cell persistence.

Within the tumor microenvironment (TME), CAR-T cells encounter a network of immune regulatory molecules and immunosuppressive cells, including dendritic cells and macrophages, which contribute to their exhaustion. Several molecular factors involved in CAR-T exhaustion have been identified: DNA methyltransferase 3A (DNMT3A) appears to contribute to CAR-T dysfunction, while ID3 and SOX4 may help counteract this process [46,47].

In the setting of neoplasia, two key strategies have been explored. The first involves modifying CAR-T cells to secrete cytokines such as IL-7 and IL-15 [48,49]. IL-7 enhances CAR-T cell proliferation and survival, while IL-15 plays a critical role in sustaining long-lasting CD8+ memory T cells. The second strategy targets the PD-1/PD-L1 axis; CAR-T cells engineered to express an autocrine PD-L1 scFv antibody have shown improved anti-tumor activity [50,51].

Although these strategies of mitigating CAR-T exhaustion are promising, they remain under investigation in preclinical models and early-phase clinical trials.

### 4.2. Clinical Applications of CAR-T Against Solid Tumors

In Table 4, the most frequently investigated antigenic targets for the design of CAR-T cells in solid tumors are reported.

#### 4.2.1. HER2 CAR-T

Human Epidermal Growth Factor Receptor-2 (HER2) is commonly overexpressed in various tumors, driving cell proliferation and tumorigenesis [52]. Given its crucial role in the development of cancers such as breast, colorectal, brain and lung cancers, HER2-targeted CAR-T cells were rapidly developed and implemented [53]. An innovative approach combines CAR-T therapy with the intratumoral administration of an oncolytic adenovirus. The viral infection of tumor cells is expected to amplify CAR-T cell activation and facilitate the destruction of cancerous cells (NCT03740256). Another noteworthy trial is exploring the localized delivery of memory-enriched HER2 CAR-T cells to treat brain and/or leptomeningeal metastases from HER2-positive cancers (NCT03696030).

#### 4.2.2. EGFR-CAR-T

The epidermal growth factor receptor (EGFR) plays a pivotal role in the development and progression of solid tumors, making it a crucial therapeutic target in various cancer types, including non-small-cell lung carcinoma (NSCLC), breast cancer, gastroesophageal cancer and colorectal cancer [54]. A phase I clinical trial (NCT01869166) of EGFR CAR-T therapy in 11 patients with EGFR+ refractory/relapsed NSCLC demonstrated partial responses in 2 patients and stable disease in 5, with a duration of 2 to 8 months, without significant toxicity [55]. Similarly, a phase I trial of EGFRvIII CAR-T cells in 10 patients with recurrent EGFRvIII+ glioblastoma (NCT02209376) showed promising anti-tumor effects, achieving a median overall survival of 8 months. Ongoing studies include three phase I and three phase I/II trials investigating novel strategies to enhance CAR-T cell efficacy, such as the CRISPR/Cas9-mediated knockout of TGF-β receptor II (NCT04976218), bi-specific EGFR/B7-H3 CAR-T cells (NCT05341492) and EGFR/CD19 dual CAR-T cells (NCT03618381). Additionally, three phase I/II trials are exploring the combination of EGFR CAR-T cells with checkpoint inhibitors targeting PD-L1/CTLA-4 (NCT03182816, NCT02873390, NCT02862028), underscoring the potential benefits of integrating checkpoint blockade with CAR-T therapy.

#### 4.2.3. CEA-CAR-T

Carcinoembryonic antigen (CEA) is a glycoprotein typically expressed during fetal development and absent in adults [56]. Elevated CEA levels are a well-established poor prognostic marker in various solid tumors, including lung [57], breast [58], pancreatic and gastric cancers [59,60]. CEA is overexpressed in more than 90% of colon cancers and approximately 50% of breast cancers [61]. Over 10 phase I/II clinical trials have explored the role of CEA-targeted therapies. Many of these trials involved the intraperitoneal administration of CEA-CAR-T cells, demonstrating good tolerability and promising therapeutic responses.

#### 4.2.4. MSLN-CAR-T

Mesothelin (MSLN) is a glycosylated, phosphatidylinositol-anchored protein predominantly expressed on mesothelial cells of the peritoneum, pleura and pericardium [62]. Several tumor cells overexpressed MSLN, making it an attractive target for CAR-T therapy. From the first clinical experience of using MSLN CAR-T emerged the efficacy but also the limitations of this construct [63]. The persistence of MSLN CAR-T is limited, inducing the discovery of novel strategies. One approach consisted of the expression of CD40L, an activation of the immune system (NCT05693844) or a combination with immune checkpoint blockades.

#### 4.2.5. ROR1

The orphan tyrosine kinase receptor ROR1 is expressed on lymphatic and epithelial malignancies such as lung and breast cancers [64]. It is highly expressed in the embryonic phase, while its expression decreases in adult cells. Several malignant cells expressed ROR1, making it a possible target for CAR-T cells [65].

#### 4.2.6. Gangliosides

Gangliosides are glycosphingolipids primarily found in neural tissue and overexpressed in certain tumors. Disialogangliosides (complex gangliosides) are linked to tumorigenesis, though their expression diminishes in non-neural healthy adult cells, where they are critical during development [66]. In contrast, mono-sialylgangliosides are believed to suppress tumor phenotypes. GD2 and GD3, two disialogangliosides, are notably overexpressed in cancers such as melanoma, sarcoma, glioblastoma, neuroblastoma, breast cancer, pediatric T cell lymphomas and lung cancer [67]. Anti-GD2 monoclonal antibodies have significantly advanced cancer immunotherapy, particularly against neuroblastoma and melanoma, due to their high expression of GD2, a glycosphingolipid with sialic acid residues, abundant in neuroectodermal tumors [50] (Ohkawa). In a phase I/II trial (NCT03373097) of anti-GD2 CAR-T cells incorporating an inducible caspase 9 suicide gene (GD2-CART01) for high-risk neuroblastoma, mild cytokine release syndrome occurred in 74% of patients, with one instance requiring suicide gene activation to eliminate CAR-T cells. The CAR-T cells demonstrated in vivo proliferation and persistence in 26 of 27 patients, with a median duration of 3 months (range: 1–30 months). The treatment achieved an overall response rate of 63%, including complete responses in nine patients and partial responses in eight [68].

### 4.3. Mucin Glican

Mucins are large, heavily glycosylated glycoproteins that form the primary component of mucus. Their structure predominantly features O-glycosylation and N-glycosylation, which are crucial for their biological functions [69]. MUC1, a membrane-bound mucin, is the most extensively studied tumor-associated mucin due to its role in cancers of glandular epithelial origin. Aberrant features such as overexpression, loss of polarization and altered glycosylation are commonly observed in adenocarcinomas, including lung, liver, colon, breast, pancreatic and ovarian cancers [70]. Several ongoing trials are testing the anti-tumor activity of MUC1-CAR-T; most of them adopted combination strategies with immune checkpoint inhibitors [71].

### 4.4. Heparan Sulfate Proteoglycans (HSPGs)

HSPGs have a pleiotropic effect on the adhesion and migration of tumor growth and also maintain cellular integrity [72]. Different types of tumors—pancreatic, breast and glioblastoma—overexpress these molecules [73]. HSPGs are composed of a core protein, and glypicans (GPCs) are cell-surface linked to membrane phospholipid phosphatidylinositol (GPI). Based on different core proteins, several HSPG families have been identified [74]. The expression of HSPGs is tumor specific; GPC3 has been found in hepatocellular and urothelial carcinoma, while GPC4 has been found in pancreatic and colorectal cancer [75,76]. This evidence led to the development of CAR-T cell therapy directed against glypicans. Preliminary results suggest that these results show promising anti-tumor activity.

### 4.5. CAR-NK

Although CAR-T therapy for solid tumors is still in the developmental stage, its potential benefits are significant and ongoing research is focused on optimizing outcomes through the use of novel constructs. Among these, CAR-NK cells and CAR-macrophages (CAR-M) have been actively explored.

NK cells represent a valid alternative to CAR-T due to their physiological role in immunotherapy against cancer cells [77]. They represent innate immune cells and are CD3-negative and CD56-positive. NK cells act without previous memory and recognize tumor cells regardless of tumor antigens. Their cytotoxic activity is regulated by a balance between activating and inhibitory receptors. Activating receptors such as DNAX accessory molecule-1 (DNAM-1), NKG2D, NKp46, NKp44 and NKp30 lead NK cells to kill tumor cells by secreting perforin and granzyme B. Fas ligand (FasL) and/or TRAIL are responsible for activating the death receptor pathway [78]. However, NK cells can also activate antibody-dependent cellular cytotoxicity (ADCC). The NK inhibitory pathway depends on the interaction between killer cell Ig-like receptors (KIRs) and major histocompatibility complex class I. Among the advantages of a cell therapy based on NK cells are the multiple sources (peripheral and cord blood, induced pluripotent stem cells and cell lines) and the good safety profile, with low incidence of CRS, neurotoxicity and GvHD [62]. Indeed, CAR-NK cells are also available in the allogeneic construct in the “off-the-shelf” therapy [79]. On the other hand, the main limitations of NK-CAR are represented by the difficulty infiltrating tumor mass due to immunosuppressive TME, the limited ex vivo expansion and the low persistence in the immunosuppressive TME. Indeed, CAR-NK showed low CAR transduction efficiency. To overcome these issues, non-viral electroporation (mRNA, transposon) transduction was used to induce CAR-NK cells to express chemokine receptors and target immunosuppressive molecules. CAR-NK cells have also been engineered to express IL-12, IL-15 and IL-18 to enhance ex vivo expansion and in vivo persistence [80]. Autologous CAR-NK have been evaluated in several clinical trials with different targets: PSMA in prostate cancer (NCT03692663), MSLN in ovarian cancer (NCT03692637) and HER2 in glioblastoma (NCT03383978). Indeed, in January 2022, the FA approved the clinical application of FT536, a CAR-NK for advanced solid tumors derived from induced pluripotent stem cells. This strategy allows for multiple CAR-NK administrations, which can be “off-the shelf” for the patients [81].

### 4.6. CAR-M

Another potential cell for CAR therapy is the macrophage. The macrophages practiced the innate immune response through phagocytic activity and the secretion of cytokines and chemokines [82]. Macrophages can be classified into M1 and M2 types based on their phenotype and functional characteristics [83]. M1 macrophages play a role in anti-tumor activity, while the M2 subtype promotes metastasis, survival and tumor growth [84]. Several studies have demonstrated that the polarization from M1 to M2 phenotype depends on the stage of the tumor: proinflammatory in the early stage and pre-cancer cells in the advanced state [85]. It is also well known that the development of tumor mass is also promoted by tumor-associated macrophages (TAMs), which are mostly M2 macrophages [86]. TAMs play a fundamental role in cancers and represent potential targets for cellular therapy with CAR-T. To overcome these obstacles, the researchers developed CAR-M constructs with M1 characteristics able to phagocyte tumor cells. Indeed, CAR-M cells stimulate the adaptive immune system to produce synergistic anti-tumor effects through antigen presentation and the activation of T cell cytotoxicity. Similarly to CAR-NK cells, CAR-M can be derived from various sources. Currently, a limited number of clinical trials are underway, showing promising efficacy and a favorable safety profile. HER2 is the most commonly targeted antigen in CAR-M therapies, although conclusive results are yet to be reported.

## 5. CAR-T Cells Beyond Oncology: Infectious Disease

### 5.1. Infectious Disease

HIV is the etiological agent of acquired immunodeficiency syndrome (AIDS), a condition marked by the depletion of CD4+T cells [87]. Although combined antiretroviral therapy (ART) has been highly successful, a definitive cure for HIV through pharmacological agents remains elusive. Maldini et al. [88] designed dual CAR-T cells that co-express second-generation CARs based on CD28 and 4-1BB signaling domains. These CARs incorporate the CD4 extracellular domain and are paired with the C34-CXCR4 fusion inhibitor to protect CAR-T cells from HIV infection. In vivo, these engineered CAR-T cells demonstrated enhanced effector functions compared to traditional second- and third-generation CARs. However, this study failed to demonstrate a decrease in viremia in the absence of ART.

Epstein–Barr Virus (EBV) is an opportunistic pathogen that establishes life-long asymptomatic infection in up to 95% of the population [89] and is associated with EBV-related B-cell cancers and post-transplant lymphoproliferative diseases. In recent years, different cellular strategies have emerged as improving efficacy in the treatment of EBV-related diseases [90]. Dragon et al. developed TCR-like CAR TRUCKs—second- and third-generation CAR T cells with inducible IL-12 expression—that target EBV antigen 3C peptide presented on HLA-B*35:01 [91]. These cells demonstrated robust activation, cytokine release and cytotoxicity against target cells. However, the ability of EBV to downregulate HLA limits its efficacy in vivo. To address this issue, Slabik et al. engineered CAR T cells targeting gp350, an antigen expressed in lytic and some latently infected EBV cells [92]. These CAR T cells effectively lysed target cells, reduced viral load and delayed tumor development in humanized mice, showing minimal side effects and no significant impact on body weight.

The herpesvirus human cytomegalovirus (HCMV) poses a significant risk of opportunistic infection in immunocompromised individuals, particularly after solid organ or hematopoietic stem cell transplantation, and can cause severe complications during pregnancy, including congenital infections [93]. CAR T cells designed to target glycoprotein B (gB) or other HCMV glycoproteins demonstrated specific cytokine release and target cell lysis in vitro, but their efficacy remains limited [94,95].

Hepatitis C virus (HCV) is a major cause of chronic liver disease and transplantation, with some patients remaining uncured despite direct-acting antivirals. Sautto et al. [96] developed scFv-based CAR T cells targeting the HCV E2 glycoprotein, showing cytotoxicity and IFN-γ secretion against E2-expressing cells and HCV-infected cells in vitro.

Hepatitis B virus (HBV) causes chronic liver infection and is a major contributor to liver cirrhosis and hepatocellular carcinoma, with 3.5% of the global population estimated to be chronically infected despite vaccines and antiviral therapies. Bohne et al. developed CAR T cells targeting HBV surface antigens, which effectively secreted IFN-γ and IL-2, lysed infected hepatocytes and eliminated cccDNA-positive cells in vitro [97]. In transgenic HBV mouse models, these CAR T cells improved viral control with minimal liver damage but failed to achieve a complete clearance, as the cells were rejected over time. Subsequent studies demonstrated that combining signaling-defective and signaling-capable CAR T cells with sublethal irradiation improved CAR T cell persistence and enhanced viral control [98].

Severe acute respiratory syndrome coronavirus 2 (SARS-CoV-2), the cause of COVID-19, has led to significant global health and economic challenges since late 2019 [99]. In addition to vaccines and drugs, immunotherapies are being explored due to severe immune dysregulation in critical cases. Guo et al. developed CAR T cells targeting the SARS-CoV-2 spike receptor, demonstrating IFN-γ secretion and the upregulation of CD69, granzyme B and perforin in vitro. In vivo, these cells reduced S1-expressing cell populations [100]. However, as COVID-19 is an acute infection, traditional CAR T cell therapy faces challenges due to lengthy manufacturing times, suggesting a need for off-the-shelf CAR T solutions [101].

### 5.2. Autoimmune Disease

Treating autoimmune diseases often requires lifelong immunosuppression, prompting researchers to explore innovative approaches to mitigate autoimmunity and achieve drug-free remission. Monoclonal antibodies, such as Rituximab, have been used to target and deplete autoreactive B-cells, showing an improvement in patient outcomes. CAR-T cells, commonly used to target CD19 and deplete B lymphocytes in B-cell malignancies, are emerging as a promising therapy for autoimmune diseases. Müller et al. reported on the outcomes of CAR T cell therapy in 15 patients with severe autoimmune diseases, including systemic lupus erythematosus (SLE, eight patients), inflammatory myositis (three patients) and systemic sclerosis (four patients), with a median follow-up of 15 months. The results were highly encouraging: all SLE patients achieved Definition of Remission in SLE (DORIS), patients with systemic sclerosis showed reduced disease activity and those with inflammatory myositis experienced significant clinical improvement. Remarkably, all 15 patients discontinued immunosuppressive therapy after CAR T infusion, maintaining negative serological markers even after B-cell recovery (median recovery time of 100 days). Regarding short-term safety, no high-grade CRS or ICANS was observed. CRS occurred in 11 patients (10 with Grade 1 and 1 with Grade 2), while ICANS was reported in only 1 patient (Grade 1) and was resolved with corticosteroids. Long-term safety was also favorable, with only one patient experiencing a severe infection (pneumonia requiring hospitalization). Other infections were mild or moderate, primarily affecting the upper respiratory tract. All 15 patients remained alive and in disease-free remission at the 15-month follow-up [101,102,103,104].

Although this study included a small patient cohort, these findings highlight CAR T cell therapy as a safe and effective option for drug-free remission in refractory autoimmune diseases. Further studies are necessary to confirm these results and integrate this approach into routine clinical practice.

## 6. Conclusions

CAR-T cells currently represent the most innovative form of cellular therapy and have revolutionized the treatment of certain B-cell hematological malignancies, integrating principles of targeted therapy, immunotherapy and gene therapy. Their potential to eradicate pathological cells has also driven their application in solid tumors and other chronic diseases. However, despite promising outcomes, the results remain inconclusive. Several limitations in effectively targeting neoplastic cells hinder the success of this therapy.

Unlike B-cell hematological malignancies, where research has focused on identifying a highly effective target, solid tumor cells not only exhibit diverse potential targets, but also exist within a complex network of cells that constitute the tumor microenvironment. This complexity has led to the development of various CAR-T designs and combination therapies (antibodies targeting immunosuppressive pathways) or the use of different cells (CAR-M or CAR-NK).

In the treatment of B-cell lymphomas, one of the primary obstacles to therapeutic success has been toxicity related to off-target effects, such as CRS and ICANS. Conversely, in solid tumors, as well as in infectious and autoimmune diseases, the key to identifying a successful target or a combination therapy remains elusive.

Subsequent prospective and randomized studies, incorporating novel molecules and constructs, will be essential to establish efficacy and achieve favorable outcomes.

## Figures and Tables

**Figure 1 cells-14-00041-f001:**
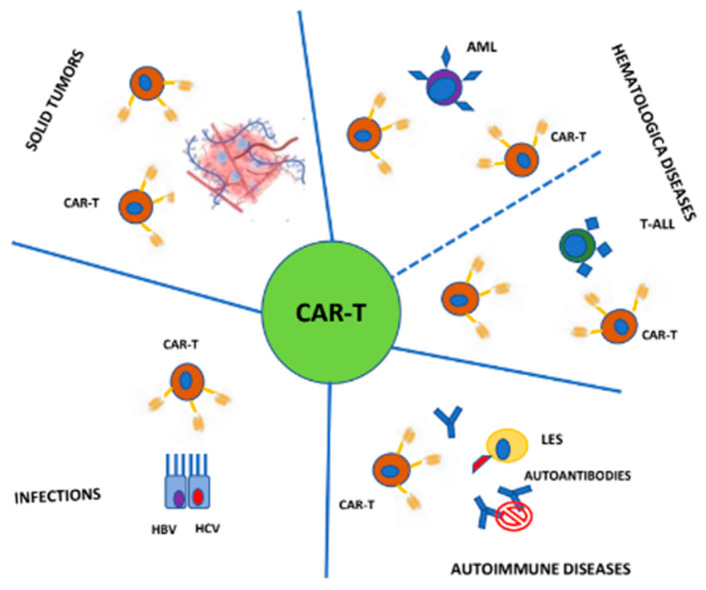
Different CAR-T clinical applications.

**Table 1 cells-14-00041-t001:** Limitations and strategies to overcome them for CAR-T in different fields.

CAR-T Setting	Limitations	Strategies
AML	Lack of antigen specificity	CAR-T cells targeting two or more antigens UniCAR-T
LAL-T	Fratricidal effect	CRISPR-Cas9 gene editing
SOLID TUMORS	Difficulty in tumor trafficking and infiltration TME T cell exhaustion	CAR-T cells expressing chemokine receptors CAR-T against Treg, TAMs and MDSCs, CAR NK, CAR M CAR-T cells to secrete cytokines such as IL-7 and IL-15 or strategy targets the PD-1/PD-L1 axis

AML: acute myeloid leukemia; LAL-T: T cell acute lymphoblastic leukemia; TME: tumor microenvironment; Treg: regulatory T-cells; MDSC: myeloid-derived suppressor cells.

**Table 2 cells-14-00041-t002:** Ongoing clinical trials CAR-T in AML.

Target	NCT	Phase	Year
CD33	NCT03971799	1–2	2020
	NCT05672147	1	2023
CD123	NCT05949125	1	2024
	NCT04230265	1	2020
CCL1	NCT04219163	1	2020
	NCT04923919	Early Phase 1	2021
FLT3	NCT05023707	1–2	2021
	NCT04884984	1–2	2017

**Table 3 cells-14-00041-t003:** Ongoing clinical trials CAR-T in T-ALL.

Target	NCT	Phase	Year
CD7	NCT06064903	1–2	2024
	NCT05043571	1	2021
	NCT05290155	1	2022
	NCT06316427	1–2	2024
	NCT05620680	Not Applicable	2021
	NCT06136364	1	2023
CD4	NCT03829540	1	2020
	NCT06197672	1	2024
CD1a	NCT05745181	1	2023

**Table 4 cells-14-00041-t004:** Ongoing clinical trials in solid tumors.

Target	Type of Cancers	NCT	Year	Phase
HER2	Malignant Neoplasm HER2-positive Breast Cancer Breast Cancer Metastatic Malignant Neoplasm in the Brain Metastatic Malignant Neoplasm in the Leptomeninges Lung Cancer HER2-positive Malignancies Advanced Solid Tumor Cancer of the Salivary Gland Bladder Cancer Breast Cancer Colorectal Cancer Ependymoma HER-2 Protein Overexpression Glioma	NCT03696030 NCT03198052 NCT04684459 NCT06241456 NCT03740256 NCT04903080 NCT04995003 NCT05768880	2018 2017 2021 2024 2020 2022 2021 2023	1 1 Early Phase 1 1 1 1 1 1
EGFR	EGFR Overexpression Lung Cancer EGFR/B7H3-positive Advanced Lung and Breast Cancer EGFR/B7H3-positive Advanced Lung Cancer Non Small Cell Lung Cancer Cancer of the Salivary Gland Bladder Cancer Breast Cancer Colorectal Cancer	NCT04976218 NCT03198052 NCT05341492 NCT06186401 NCT05060796 NCT03740256	2022 2017 2022 2024 2024 2020	1 1 Early Phase 1 1 Early Phase 1 1
CEA	Bile Duct Cancer Breast Cancer Colorectal Cancer Esophagus Cancer Breast Cancer Colon Cancer Esophageal Cancer Gastric Cancer Breast Cancer Cholangiocarcinoma Colon Cancer Esophagus Cancer Colon Cancer Esophageal Cancer Gastric Cancer Pancreatic Cancer Rectal Cancer Colorectal Cancer Esophageal Cancer Metastatic Tumor Pancreatic Cancer Breast Cancer Cholangiocarcinoma Colon Cancer Esophagus Cancer Colorectal Cancer Esophageal Cancer Metastatic Tumor Pancreatic Cancer Colorectal Cancer Metastatic Liver Cancer Cancer Colorectal Neoplasms Colorectal Cancer	NCT06043466 NCT06006390 NCT06126406 NCT05538195 NCT05415475 NCT06010862 NCT05396300 NCT05240950 NCT05736731	2023 2023 2023 2022 2021 2023 2022 2022 2023	1 1/2 1 1/2 1 1 1 1 ½
MSLN	Colorectal Cancer Advanced or Metastatic Solid Tumors Solid Tumor Mesothelin-positive Advanced Malignant Solid Tumors Different solid Tumors MSLN+ Mesothelin Pancreas Cancer Lung Cancer Colon Cancer Colorectal Cancer Liver Cancer Lung Cancer	NCT05089266 NCT05693844 NCT06248697 NCT05848999 NCT05166070 NCT05779917 NCT03198052 NCT06051695 NCT06196294	2021 2023 2023 2023 2022 2023 2017 2024 2024	1 1/2 Early Phase 1 1 Early Phase 1 1 1 1/2 1
ROR1	Different Solid Tumors	NCT05748938	2023	1/2
GD2	Different Solid Tumors Different Solid Tumors Different Solid Tumors Ewing Sarcoma Neuroblastoma Neuroblastoma Recurrent Embryonal Tumor Ependymal Tumor Brain Tumors Neuroblastoma Osteosarcoma, Neuroblastoma Lung Cancer Non-Small Cell Lung Cancer Small Cell Lung Carcinoma Neuroblastoma Neuroblastoma Osteosarcoma Glioma of Brainstem Glioma of Spinal Cord	NCT05437315 NCT05438368 NCT05437328 NCT03373097 NCT04099797 NCT05298995 NCT06684639 NCT04539366 NCT05620342 NCT03294954 NCT03721068 NCT04196413	2022 2022 2022 2022 2020 2023 2024 2022 2023 2018 2019 2020	1/2 1/2 1/2 1/2 1 1 Early Phase 1 1 Early Phase 1 1 1 1
MUC1	Lung Cancer Metastatic Breast Cancer	NCT03198052 NCT04020575	2017 2020	1 1
GPC3	Mesothelin Pancreas Cancer Liver Cancer Lung Cancer Hepatocellular Carcinoma Lung Cancer Hepatocellular Cancer Hepatocellular Carcinoma Metastatic Hepatocellular Carcinoma Advanced Hepatocellular Carcinoma Liposarcoma Liver Cancer Malignant Rhabdoid Tumor Rhabdomyosarcoma Hepatocellular Carcinoma Liver Cancer Malignant Rhabdoid Tumor Rhabdomyosarcoma Embryonal Sarcoma of the Liver Liposarcoma Liver Cell Carcinoma Malignant Rhabdoid Tumor	NCT05779917 NCT06196294 NCT03198546 NCT03198052 NCT05003895 NCT06461624 NCT04377932 NCT06084884 NCT04715191 NCT05103631	2023 2024 2017 2017 2021 2024 2021 2023 2024 2021	1 1 1 1 1 1 1 1/2 1 1

## Data Availability

Not applicable.

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
