# Peer review of "CAR-T Therapy Beyond B-Cell Hematological Malignancies"

_cells, 2025, doi:10.3390/cells14010041_

Round 1

Reviewer 1 Report

Comments and Suggestions for Authors

This review focused on exploring the prospect of CAR T cell therapy beyong B cell malignancies. Overall article is well written following are some comments. 

1. There are a lot of improvements which can be done for this manuscript, for instance the authors should provide my descriptive details of the ongoing clinical trial in table 1 and table 2. It will be difficult for the readers to decipher about the type of clinical trial just by a number. 

2. Author should also include a summary figure to make that a reference point for an easy read. 

Comments on the Quality of English Language

There are substantial scope to improve the english language in this manuscript to make it easy to read 

Author Response

  1. Thank you for your suggestions, which have been greatly appreciated. You are absolutely right in stating that the topic is particularly wide. The topic has been deliberately narrowed to summarize the diverse and emerging applications of CAR T cells. The complexity of this subject has limited our ability to delve deeply into the details of the clinical studies, which, while promising, remain preliminary. However, we have emphasized the limitations of CAR T application in the three non-B hematologic domains, which differ significantly from those encountered in B-cell pathology.
  2. To simplify and guide the reader, we have added a table and a figure, as per your suggestion, and made adjustments to the main text. We hope these modifications improve the clarity and accessibility of the review. Thank you again for your valuable feedback

Reviewer 2 Report

Comments and Suggestions for Authors

The review does not add new insights beyond what is already available in the literature. It reads as a mere compilation of studies and trials, much like what one could easily find on clinicaltrials.gov. Summarizing each target antigen in just 3-4 lines oversimplifies years of extensive preclinical and clinical research, diminishing the potential depth and value of the review. Attempting to cover hematological malignancies solid tumors and autoimmune diseases may be overly ambitious for the scope of this review, leading to a superficial and simplified description of clinical studies. The lack of detailed results further limits the review's value and contribution to the field. The authors should consider narrowing their focus to one of the two areas. Alternatively, they could adopt a more critical perspective, emphasizing the key challenges and limitations faced by CAR T cell therapies beyond B-cell malignancies, as they did in the second part of the review. An alternative and innovative perspective could be a critical evaluation of CAR T cell therapy applications in autoimmune diseases, highlighting differences in study design, starting sources for CAR T generation, and patient management and follow-up as compared to the setting of CAR T cell therapy for cancer.

Author Response

We fully agree with your observations, and we sincerely thank you for your well-structured and thoughtful comments. Your feedback has helped us identify areas that could be improved to enhance the clarity and accessibility of our review.

You are correct in noting that the topic is somewhat ambitious, as it encompasses the broad application of CAR T cells beyond B-cell malignancies. When proposing this topic to the managing editor, we emphasized that the intent of this review was to highlight how CAR T therapy has the potential to revolutionize the treatment of other pathologies. In doing so, we presented results from clinical studies that, while promising and encouraging, remain preliminary.

Our primary aim was to underscore the unique limitations of applying CAR T therapy in these contexts, which differ significantly from those encountered in B-cell hematological diseases. These limitations may explain the slower progress in achieving widespread implementation of CAR T therapy in solid tumors, for example.

To address your suggestions, we have added a table and a figure to clarify these aspects and revised portions of the text accordingly. We hope that these changes, based on your valuable input, have made the review more accessible and comprehensible. Thank you again for your insightful feedback